# Technical Note: Extending the SWAT2012 and SWAT+ models to simulate pesticide plant uptake processes

Hendrik Rathiens<sup>1</sup>, Jens Kiesel<sup>1</sup>, Jeffrey Arnold<sup>2</sup>, Gerald Reinken<sup>3</sup>, Robin Sur<sup>3</sup>

- <sup>1</sup>Stone Environmental, 535 Stone Cutters Way, 05602 Montpelier (VT), USA <sup>2</sup>USDA-ARS, Grassland Soil and Water Research Laboratory, 808 East Blackland Rd., 76502 Temple (TX), USA <sup>3</sup>Bayer AG, Research & Development Crop Science, Environmental Safety Ass. & Strategy, Building 6692 2.14, 40789 Monheim, Germany
- Correspondence to: Jens Kiesel (jkiesel@stone-env.com)

**Abstract.** The SWAT model is widely used for simulating pesticide fate and transport in agricultural watersheds but currently lacks the ability to represent chemical uptake by plants, which is a significant pathway particularly relevant for stable compounds that can persist in the root zone. To address this limitation, the publicly available SWAT code was modified to incorporate pesticide plant uptake processes, building upon recent improvements in chemical subsurface transport pathways. The implementation calculates chemical plant uptake based on plant water uptake, substance-specific uptake factors, and concentrations of the chemical in soil pore water. The enhanced model was tested in two agricultural catchments using a stable pesticide soil metabolite with known plant uptake characteristics. Results demonstrate that including plant uptake processes reduced metabolite concentrations in streamflow by 5-17%. The implementation reveals the importance of plant uptake as a sink, particularly for persistent compounds, and provides new capabilities for assessing agricultural pesticide management practices or mitigation strategies and their effects on environmental fate. The functionality has been implemented in both SWAT2012 and SWAT+, with code provided as an electronic supplement to this technical note.

#### 1 Introduction

Pesticide modelling at the watershed scale has become essential for understanding pesticide fate and transport in the environment. Models like the Soil and Water Assessment Tool (SWAT; Arnold et al., 1998; Bieger et al., 2017) enable analysis of various scenarios including management practices and mitigation strategies to reduce pesticide concentrations in water bodies (Holvoet et al., 2007). While SWAT has been successfully applied worldwide for simulating pesticide transport (Gassman et al., 2014) and was recently extended to include pesticide transport through tile drains and groundwater (Rathjens et al., 2023), some key processes remain unaccounted for. One such process is pesticide plant uptake, which represents a significant pathway in the environmental fate of pesticides, particularly for soil metabolites that can persist in the root zone. Accurately representing this pathway is challenging since, as Fantke et al. (2013) describe, the dynamics of substance masses

in multi-compartment plant-environment systems are controlled by both fate processes of chemicals and functions describing substance application or emission. Moreover, fate processes in field crops (i.e., uptake, translocation and elimination mechanisms), depend on substance properties and vary between individual plant species.

Empirical studies have shown that plant uptake of pesticides can vary substantially, removing between 2% and 98% of soil water pesticide concentrations (Lamshoeft et al., 2018). The transport of compounds into plant cells can occur via three main pathways: apoplastic (moving between cells along cell walls), symplastic (moving through cells via plasmodesmata) and transmembrane (moving through cells via cell membranes), with the ability to cross membranes being determined by the physicochemical properties of the compound (Fantke et al., 2013; Schriever and Lamshoeft, 2020). This uptake process is primarily driven by plant water uptake via the xylem, with the accumulation of pesticide mass in plants showing a linear relationship to water uptake (Lamshoeft et al., 2018). Regarding pesticides, the process is particularly relevant for stable metabolites that can accumulate in soil and subsequently be taken up by plants, potentially affecting their environmental fate and transport pathways. While other established chemical exposure models like RZWQM (Hanson et al., 1998), PEARL (van den Berg et al., 2016), and PRZM (Carsel et al., 1985) incorporate pesticide plant uptake processes, this pathway has not yet been accounted for in SWAT, potentially leading to uncertainties in fate predictions, especially for compounds with significant plant uptake potential.

The chemical plant uptake process is represented in environmental fate models through the Plant Uptake Factor (PUF), which acts as a resistance term determining the fraction of dissolved pesticide in soil water that is taken up with the transpiration stream. While plant uptake varies with both plant species and pesticide properties, regulatory models typically implement PUF as a single pesticide-specific parameter due to limited availability of plant species-specific data. PUF values range from 0 (no uptake) to 1 (complete uptake with transpired water), with most substances having a value between these extremes. Consistent with current implementations (e.g., PEARL), the integration of plant uptake into SWAT assumes that only pesticides in the soil solution are available for uptake, while sorbed fractions must first desorb before becoming available for plant uptake.

50

This technical note presents the implementation of chemical plant uptake into the SWAT-2012 model (version 681) and its successor SWAT+ (version 61.0), collectively referred to as SWAT. Building upon recent improvements in pesticide transport simulation through subsurface pathways (Rathjens et al., 2023), the new functionality calculates pesticide uptake based on plant water uptake, a substance-specific uptake factor, and soil pore water pesticide concentration. The implementation considers key factors such as rooting depth, vertical water uptake distribution, and the relationship between plant growth and water availability. The functionality is evaluated in two agricultural catchments using a stable pesticide soil metabolite with known plant uptake characteristics. By incorporating this process, the capability of SWAT to simulate pesticide fate and transport is enhanced, particularly for substances where plant uptake represents a significant removal pathway from soil, including persistent metabolites that can accumulate in the root zone.

## 2. Software description

# 2.1. SWAT model structure and pesticide processes

SWAT is a semi-distributed model that simulates water, sediment, nutrient, and chemical fluxes at multiple scales throughout a watershed. The model divides catchments into subbasins based on stream density, stream confluences, and user-defined outlet locations. Each subbasin is further subdivided into hydrologic response units (HRUs) representing unique combinations of land use, soil, and slope classes. HRUs operate as independent computational units with distinct parameterizations and management practices.

Within each HRU, SWAT simulates various hydrological processes including surface runoff, infiltration, evaporation, plant water uptake, lateral flow, tile drain flow, and percolation. For chemicals/pesticides, the model accounts for multiple fate processes: wash-off from plant surfaces, degradation on foliage, and transport with surface runoff and erosion. Within the soil profile, the model simulates partitioning between solid and liquid phases in soil, biodegradation, and movement with water fluxes (lateral flow, tile flow, groundwater flow, percolation). Pesticide movement through the soil profile is determined by environmental fate properties (primarily the soil adsorption coefficient) and environmental conditions. For detailed information on the calculation of fluxes and concentrations, readers are referred to the SWAT theoretical documentation (Neitsch et al., 2011).

Two important differences between SWAT2012 and SWAT+ should be highlighted. Firstly, while SWAT2012 initially simulated chemical transport primarily through surface runoff, erosion, and lateral flow, recent model developments (Rathjens et al., 2023) have added transport capabilities through tile drains and groundwater. For the plant uptake implementation in SWAT2012, this study builds upon the version introduced in Rathiens et al. (2023). Secondly, the formation of metabolites from parent compounds is not directly implemented in SWAT2012, requiring separate calculation and implementation using pseudo chemical applications. In SWAT2012, metabolite formation must be pre-calculated based on the soil degradation halflife and formation fraction of the parent compound, with the metabolite then applied as a pseudo pesticide application at an estimated soil depth and appropriate timing to represent the expected formation pattern. Regarding SWAT+, in the current version both processes (metabolite formation using first-order decay and chemical transport through all hydrological pathways) are already implemented along with several improvements in pesticide fate modeling such as a more detailed representation of landscape units and their connections and enhanced flexibility in defining agricultural management operations using decision tables (Rathjens et al., 2022). Metabolite formation occurs continuously within each soil layer through first-order degradation kinetics, with the timing controlled by the half-life of the parent compound. The formation of metabolites from parent compounds is controlled by a pesticide-specific formation fraction parameter, allowing metabolites to form at different depths throughout the soil profile depending on the vertical distribution of the parent compound. This depth-distributed formation is important for accurately representing metabolite fate, as metabolites formed in deeper soil layers may have different transport pathways and plant availability compared to those formed near the surface. However, neither SWAT2012 nor SWAT+ include pesticide plant uptake processes, which is particularly important for water-soluble compounds and stable metabolites that can accumulate in the root zone.

# 2.2. Description of the plant uptake functionality






Plant uptake represents a significant pathway for pesticide removal from soil (Lamshoeft et al., 2018), primarily driven by plant water uptake, with pesticide accumulation in plants showing a linear relationship to water uptake rates. The pesticide uptake calculation builds on the existing plant water uptake functionality in SWAT, which simulates the vertical distribution of root water uptake through the soil profile.

First, the model simulates dynamic root growth for annual crops based on accumulated heat units, while using maximum rooting depth for perennial vegetation. Note that the variable names used in the following equations follow the SWAT2012 Fortran code to enable comparison with the source code, even though some may not be immediately intuitive. The potential water uptake (*sum*) from the soil profile follows an exponential distribution with depth, reflecting the typically observed higher root density near the soil surface:

$$sum = ep_{max} \cdot \left(1 - \exp\left(-ubw \cdot \frac{gx}{sol_{rd}}\right)\right) \cdot \frac{1}{uobw}$$

where  $ep_{max}$  is the maximum plant transpiration [mm H<sub>2</sub>O], ubw is the water uptake distribution parameter [-] (set to 10 in SWAT), gx is the depth to the bottom of the current layer [mm],  $sol_{rd}$  is the rooting depth [mm], and uobw is the uptake distribution normalization parameter [-] (set to  $1 - \exp(-ubw)$  in SWAT). This distribution, as stated in the SWAT theoretical documentation (Neitsch et al., 2009), ensures that approximately 50% of water uptake occurs in the upper 6% of the root zone, which is consistent with the observed decrease in rooting density reported by Jackson et al. (1996) and Feddes et al. (1976). It should be noted that the water uptake distribution parameters (ubw and uobw) are fixed in the current SWAT implementations and can only be modified by altering the source code, not through input files.

The potential water uptake for each soil layer is then calculated as the difference between uptake at layer boundaries and adjusted for compensation between layers:

$$wuse_k = sum - sump + (sump - xx) \cdot epc$$

where  $wuse_k$  is the water uptake for layer k [mm H<sub>2</sub>O], sump is the cumulative potential uptake to current depth [mm H<sub>2</sub>O], xx is the actual uptake from previous layers [mm H<sub>2</sub>O], and epco is the plant uptake compensation factor [-].

The actual water uptake ( $wuse_{ks}$ ) is limited by soil moisture availability using a reduction factor when soil water content falls below 25% of field capacity:

$$wuse_{ks} = \begin{cases} wuse_k \cdot \exp\left(5 \cdot \left(4 \cdot \frac{solst_k}{solfc_k} - 1\right)\right), \text{ if } solst_k < \frac{solfc_k}{4} \\ wuse_k, else \end{cases}$$

where  $solst_k$  is the soil water storage [mm H<sub>2</sub>O] and  $solfc_k$  is the field capacity water content [mm H<sub>2</sub>O] in layer k. Building on this water uptake framework, the pesticide plant uptake for each soil layer is calculated as:

where yy is the pesticide uptake from the layer [kg/ha], pstuptk is the pesticide-specific plant uptake factor PUF [-], wuse<sub>ks</sub> is the actual water uptake from soil layer k [mm H<sub>2</sub>O], and solpstconc<sub>k</sub> is the soil pore water chemical concentration [kg/mm-ha]. The parameter pstuptk corresponds to the Plant Uptake Factor (PUF) as defined in Lamshöft et al. (2018), representing the fraction of dissolved pesticide taken up with transpired water by both roots and shoots. While pstuptk is theoretically influenced by both plant and compound properties, it is implemented as a compound-specific parameter in the model. This approach reflects the limited availability of plant-specific uptake data. The implementation allows for tiered assessment approaches, where conservative default values (typically pstuuptk = 0) can be refined with experimentally determined values when available.

The implementation ensures that chemical plant uptake occurs only during active plant growth periods when sufficient water is available, that the distribution follows the established water uptake pattern with depth, and that uptake is limited by available chemical mass and concentration in each layer. In addition, the chemical mass taken up is tracked in plant tissue and removed from the soil storage. While the biological process of plant uptake encompasses multiple pathways once a chemical is absorbed, including (1) transformation into other compounds through plant metabolic processes, (2) transport to different parts of the plant, or (3) retention of chemical residues in various plant tissues, these internal plant processes are not simulated in this implementation. Instead, similar to other models such as PEARL, the uptake is represented as a one-way removal from the soil system. Once a chemical is taken up by a plant, it is considered permanently removed from the soil system and does not re-enter the soil through processes like root exudation. This is in line with current research that does not indicate that this pathway significantly contributes to the cycling of pesticides within the soil-plant system (Eze and Amuji, 2024). Similarly, and consistent with PEARL, our implementation does not explicitly simulate the decomposition of plant residues and the potential subsequent release of pesticides back into the soil if plant residues remain in the field after harvest. The conceptual approach adopted from PEARL has been scientifically validated through both theoretical development (Leistra et al., 2001) and field validation studies (Bouraoui et al., 2007). Furthermore, Jorda et al. (2021) compared the simplified passive advective uptake approach from PEARL with a mechanistic 3D root model, concluding that the simplified approach is effective for regulatory applications while acknowledging limitations in heterogeneous soils. This validated framework aligns well with our watershed-scale modeling objectives, where computational efficiency and limited data availability necessitate a simplified but effective representation of plant uptake processes.

# 2.3. Implementation in SWAT2012 and SWAT+






The implementation of pesticide plant uptake in SWAT2012 (version 681) required several code modifications. A new subroutine (*pup.f*) was added to calculate pesticide uptake for each soil layer based on water uptake patterns, pesticide concentrations, and substance-specific uptake factors. The subroutine was integrated with existing soil water balance routines and mass balance tracking was updated to account for pesticide removal via plant uptake. In addition, minor changes were

made to other subroutines for technical reasons, e.g., to produce HRU level output and to write the new parameters to output files. These changes are not discussed here but are included in the code provided with the electronic supplements.

The modified model maintains compatibility with the input files of the original SWAT2012 code. The only change required to the default SWAT2012 input parameters is the addition of the pesticide plant uptake factor in the basins.bsn input file. This parameter (PESTUPTK) must be added manually to line 138 of the basins.bsn file and has a default value of:

0.0000 | PESTUPTK: pesticide plant uptake factor - 0 no uptake, 1 complete uptake

The decision to implement the pesticide uptake parameter in the basins.bsn file for SWAT2012 was driven by structural constraints as the model only allows simulation of one pesticide at a time. Since this pesticide must be defined in basins.bsn along with other pesticide cycling parameters (such as *PERCOP*), the plant uptake parameter was added to the same file. In contrast, SWAT+ can process multiple pesticides simultaneously, allowing the uptake parameter to be implemented within the pesticide data module where it can be specified individually for each compound. A compiled Windows executable and the complete model code are provided as electronic supplements.

For SWAT+, the pesticide plant uptake functionality is already integrated into the publicly available repository since version (61.0) in subroutine *pest\_pl\_up.f90*. Figure 1 shows a schematic representation of the plant, soil, and groundwater pesticide transport processes including the newly implemented plant uptake pathway in pup.f (SWAT2012) and pest\_pl\_up.f90 (SWAT+). The pesticide leaching routines (*pestlch.f* and *pest\_lch.f90*) were modified to track chemical soil water concentrations by soil layer, providing the data required to incorporate chemical plant uptake into the existing plant water uptake processes.

#### 3. Application




The modified SWAT2012 and SWAT+ models were tested in the same two agricultural catchments in Western Europe that were previously used to evaluate pesticide transport through tile drains and groundwater (Rathjens et al., 2023). Catchment names and location as well as detailed descriptions and names of the chemicals were anonymized for this publication. The catchment characteristics are summarized in Table 1; catchment 1 (C1) was used to evaluate SWAT2012 while catchment 2 (C2) was used to evaluate SWAT+. The soil characteristics in both catchments reflect typical agricultural landscapes in 180 Western Europe. In C1 soils are mostly sandy with higher silt and clay content mostly in close proximity to the streams. C2 has heavier soils with predominantly silt-loam and sandy-loam soils. Both catchments have soil profiles extending to 1.5-2.0 m depth, with tile drains typically installed at 0.8-1.2 m depth in the agricultural areas (52% of C1 and 48% of C2 as indicated in Table 1). In both catchments, pesticide application data were available along with observations of streamflow, pesticide, and pesticide metabolite concentrations. All data sources overlap temporally from October 2016 to April 2024 for catchment 185 1 (C1) and from June 2010 to December 2013 for catchment 2 (C2). The metabolite concentration data were collected using automated time-controlled sampling in both catchments. In C1, automatic samplers collected samples that were composited into weekly mixed samples. In C2, automatic samplers collected time-based composite samples at varying frequencies (initially 4 per day, then 2 per day, and finally 1 per day) which were composited into daily samples. However, metabolite analysis frequency varied, with samples analyzed every other day in 2010-2011 and daily in 2012-2013, resulting in metabolite concentration data for 64% of the monitoring period. Data gaps exist in both time series as shown in Figure 2 and Figure 3 b and c. The pesticide is a commonly used chemical typically applied in late autumn on winter grains or in spring on corn. Based on the pesticide's half-life, it is classified as "readily degradable", its mobility is classified as "moderate", and it is considered "readily soluble" in water (FAO, 2000). In contrast, the metabolite is stable ("very slightly degradable"), "highly mobile", and "highly soluble". Since the catchment characteristics and parent pesticide behavior have been thoroughly documented in Rathjens et al. (2023), we focus here on the implementation and impact of plant uptake processes for the stable metabolite.

# 3.1. Model parameterization and calibration

The model parameterization followed standard procedures considering climate, topography, soil, and land use properties. Application data on respective crops were available with approximate amounts and timing for C1 and as field-specific applications for C2. While the previous study (Rathjens et al., 2023) was conducted in the same catchments, recalibration was necessary due to evaluating a different soil metabolite, SWAT version updates, newly available data for catchment C2, and the implementation of the plant uptake process. The calibration was conducted with the plant uptake functionality enabled with a compound-specific uptake factor of 0.305 for the metabolite, based on laboratory studies conducted by Bayer (personal communication) with agricultural crops. It is important to note that the PUF should not be treated as a calibration parameter. To maintain its physical meaning and avoid using it to compensate for other model deficiencies, PUF values should be derived from laboratory studies or literature values rather than adjusted during calibration. In this study, the experimentally-determined value of 0.305 was held constant throughout the calibration process.

The calibration strategy for both catchments involved manual parameter exploration followed by automated optimization. First, parameters and their respective ranges were identified through manual one-at-a-time sensitivity analysis based on previous studies (Rathjens et al. 2023, Rathjens et al. 2022) and experience. Nine parameters were identified in C1 and 12 parameters in C2. Then, Latin Hypercube Sampling with 12,000 and 20,000 parameter sets were conducted in C1 and C2, respectively. A multi-objective calibration was implemented using weighted criteria, with streamflow contributing 33.3% and metabolite concentrations 66.7% to the objective function. For catchment 2 (C2), the entire evaluation period (06/2010-12/2013) was used for calibration (Table 1). This is a common approach used for hydrologic and pesticide model calibration when the observed data period is relatively short (Daggupati et al., 2015). For catchment 1 (C1), separate calibration (10/2016-12/2019) and validation (01/2020-04/2024) periods were established due to the longer time period of available metabolite observations (Table 1) to assess the predictive skill of the model. For identifying the optimal parameter set, the complete time series was used for calibration according to Shen et al., (2022) in a second step. For both catchments, the top 20 model runs based on the weighted objective function were selected as final parameterizations. For C2, where multiple sources of the metabolite investigated exist beyond the simulated parent compound that are not accounted for in the model, an additional selection criterion was applied. Specifically, only parameter sets that achieved KGE (Kling et al., 2012) values >0 for both the

parent pesticide and another metabolite (that is exclusively formed by the pesticide) were considered eligible. This constraint was implemented to prevent the model selection from compensating for the expected underestimation of the metabolite under investigation through unrealistic parameter combinations.

## 3.2. Results and discussion





The evaluation of model performance focused on two key aspects: (1) the overall ability to simulate streamflow and metabolite concentrations, and (2) the specific impact of plant uptake on fate and transport of the soil metabolite. Observed and simulated streamflow and metabolite concentrations are shown in Figure 2 and Figure 3 for C1 and C2, respectively. The models demonstrated a very good hydrologic performance for both catchments, establishing a reliable foundation for evaluating chemical transport and fate processes. In catchment C1, the validation period achieved an average Kling-Gupta Efficiency (KGE) of 0.76 for streamflow, with its three components correlation (r) of 0.81, bias (β) of 1.07, and variability ratio (γ) of 0.88. Similarly strong performance was observed in C2, where the evaluation period showed comparable metrics with an average KGE of 0.78, r of 0.79, β of 1.02, and γ of 0.97. The three KGE components show that streamflow timing (indicated through r) sets the limit on the performance. A comprehensive overview of the streamflow and metabolite concentration performance metrics is provided in Table 2.

Building on this foundation, the analysis of metabolite transport and concentrations showed consistent patterns across both catchments. Metabolite concentrations in streamflow for the calibrated models (Figure 2b and 3b) showed comparable magnitudes between C1 and C2, with maximum values below  $20 \mu g/L$ . The transport dynamics across both catchments were similar, suggesting consistent underlying processes despite differences in catchment characteristics. For C2, while the metabolite dynamics were well represented, it is important to note that not all sources of the metabolite were considered in the model. The metabolite under investigation can be formed from multiple parent compounds used in the catchment or other, non-agricultural uses, but our simulation included only one parent pesticide (likely the most significant contributor) due to limited application data for other potential parent compounds. This simplification leads to an expected underestimation of simulated concentrations compared to observations, which is visible in the low bias ratio  $\beta$  of 0.61. Despite not visibly improving the match with observations in C2, the inclusion of plant uptake remains important as it represents a known physical process that affects pesticide fate. The underestimation due to missing parent compounds masks the improvement from including plant uptake, but implementing this process enhances the mechanistic representation of the model and enables more realistic mass balance calculations.

The implementation of plant uptake processes significantly improved chemical fate representation, particularly during growing seasons. In C1, the model without plant uptake (Figure 2c) tended to overestimate metabolite concentrations with a bias ratio ß of 1.2. Including plant uptake (Figure 2b) led to improved simulations, reducing average metabolite concentrations by 17% (ß of 1.03). Similarly, in C2 (Figure 3), plant uptake implementation resulted in a 5% reduction (ß from 0.66 to 0.61) in metabolite concentrations in streamflow. The difference in impact between catchments can be attributed to multiple interacting factors. While C1 has 73% agricultural land compared to 80% in C2, the crop rotations differ significantly, with C1 including

more corn and C2 being dominated by winter cereals. These crops have different growing periods and biomass development patterns, affecting the timing and magnitude of water and chemical uptake. The hydrologic regimes also differ between catchments, with runoff ratios of 28-36% in C1 versus 38-48% in C2, which can be attributed to differences in soils (Table 1). This suggests different partitioning between evapotranspiration (which drives plant uptake) and runoff/drainage pathways (which transport chemicals before plant uptake can occur). Furthermore, the proportion of agricultural land actually treated with the parent pesticide varies between catchments, affecting the spatial distribution of metabolite formation and availability for plant uptake. Different pesticide products with varying metabolite formation rates are used across the catchments, with some products in C2 potentially having reduced metabolite formation, further influencing the relative importance of plant uptake as a removal pathway.

The influence of plant uptake is visible throughout the whole year. As the plant uptake reduces metabolite pore water concentrations during the growing season, hydrologic conditions such as occurrence, timing, and magnitude of lateral and tile drain flow control the subsequent transport of the metabolite (Figure 2c and Figure 3c). To better understand these uptake dynamics, a detailed temporal analysis was conducted for an individual HRU in C1 over a 6-year crop rotation sequence (2016-2022), as presented in Figure 4. The analysis shows how biomass development, metabolite concentrations, and uptake patterns vary across different crops and seasons. Biomass development curves aligned well with expected agricultural yields. For example, corn silage achieved approximately 10 t/ha, corresponding to typical target yields when accounting for dry mass content. Plant metabolite concentrations showed characteristic patterns, with initial fluctuations during early growth stages stabilizing as biomass increased, typically reaching equilibrium concentrations of approximately 1 mg per kg biomass in main crops and 0.1 to 0.4 mg/kg in cover crops.

This temporal analysis also showed clear seasonal patterns in soil metabolite mass, varying between 20 to 100 g/ha with peaks typically occurring during winter months. Over the six-year period, the cumulative mass removed through plant uptake reached approximately 60 g/ha in agricultural areas. These findings help explain the simulated 17% and 6% reductions in average metabolite concentrations in streamflow for C1 and C2 respectively when plant uptake was implemented in the model. The implemented plant uptake factor of 0.305 (or 0.0 for no uptake) results in plants extracting proportionally more water than metabolite, leading to increased metabolite concentrations in soil water. These concentrated metabolites are subsequently transported out of the soil and into the stream via lateral and groundwater processes during wetter periods, explaining the observed seasonal and annual patterns in metabolite soil water (Figure 4) and streamflow concentrations (Figure 2b and 3b). The implementation approach for metabolite processes differs between the two model versions. SWAT2012 (used for C1) requires pseudo applications to represent metabolite formation, while SWAT+ (used for C2) directly simulates formation through first-order decay. This enhancement in SWAT+ enables more realistic representation of metabolite formation and allows for detailed investigation of formation pathways. Despite these differences in metabolite processes implementation, both versions demonstrated similar performance in simulating metabolite transport and plant uptake processes.

Several uncertainties and limitations should be considered when interpreting these results. The plant uptake factor was assumed to be constant across all crop types, although literature suggests some variation may exist (Fantke et al., 2013). Additionally,

the current implementation assumes complete removal of accumulated chemical mass in plant tissue, which may not fully represent all potential environmental fate pathways, which is consistent with other models like PEARL. While this study primarily focused on evaluating the implemented plant uptake mechanism through concentrations at the catchment outlet, supplementary analyses of metabolite dynamics across different soil types and vertical movement between layers were conducted and showed plausible behavior. However, these additional results are not discussed in detail here (beyond what was provide in Figure 4) due to limited validation data.

#### 4. Summary and conclusion




The SWAT model code was extended to incorporate chemical plant uptake processes, building upon recent developments in subsurface transport pathways. The implementation was conducted for both SWAT2012 and SWAT+ and includes new subroutines for calculating chemical uptake based on plant water uptake patterns, substance-specific uptake factors, and soil pore water concentrations. Only minor modifications to the standard SWAT input files are required, specifically the addition of a plant uptake factor parameter.

The application of the enhanced model in two agricultural catchments demonstrates the importance of including plant uptake processes when simulating persistent soil metabolites. The implementation reduced metabolite concentration in streamflow by 5-17% in various degrees over the year, showing the strong link between metabolite transport and hydrological processes. The ability to track pesticide movement through the plant uptake pathway provides valuable insights into the fate and transport of chemicals, especially for stable compounds that can accumulate in the root zone. This improved process representation supports more accurate environmental exposure assessments and enables better evaluation of agricultural pesticide management and chemical mitigation practices.

The developed functionality fills an important gap in watershed-scale pesticide modeling by using a simple parameterization approach via a single uptake factor. The code has been made available to the SWAT2012 development team for potential inclusion in future official releases. For SWAT+, the functionality is already integrated starting with version 61.0. SWAT+ also offers additional advantages over SWAT2012 through direct simulation of metabolite formation and enhanced agricultural management options and is recommended for future assessments. However, some limitations remain, such as the assumption of uniform uptake factors across crop types, the fixed water uptake distribution parameters, and the simplified handling of removing pesticide mass accumulated in the plant. While the current implementation uses a single compound-specific uptake factor due to limited data availability, future model applications could potentially incorporate crop-specific uptake factors as more experimental data becomes available, allowing for more refined predictions of plant uptake across different cropping systems. Additionally, future developments could address the return of pesticides to soil through crop residues at harvest time, though this would require substantial model enhancements to simulate pesticide release during plant residue decomposition and in-plant partitioning between harvested and residual biomass. Despite these limitations, the extended SWAT versions

provide valuable tools for risk assessors and watershed managers studying the environmental fate of pesticides, their metabolites, and other constituents.

# Code availability

The SWAT2012 source code and compiled Windows executables are available from Stone Environmental's GitHub repository (<a href="https://github.com/StoneEnv/SwatPestPlantUptake">https://github.com/StoneEnv/SwatPestPlantUptake</a>) under the GNU General Public License v3. The SWAT+ source code is available via the official SWAT+ GitHub repository (<a href="https://github.com/swat-model/swatplus">https://github.com/swat-model/swatplus</a>) under the LGPL-2.1 license.

#### 325 Author contribution

Conceptualization: HR, JK, RS; Data curation: JK, HR, RS, GR; Methodology: JK, HR; Software: JA, HR, JK; Supervision: RS; Model Calibration and Visualization: JK; Validation: JK, HR; Writing – original draft: HR; Writing – review & editing: JK, HR, JA, RS, GR.

# **Competing interests**

The authors declare no competing interests.

## Disclaimer

The authors accept no responsibility for any liability arising from the use of this manuscript, the provided source code and model.

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

Table 1: Catchment characteristics of the two anonymized catchments in Western Europe

| Catchment Characteristics                       | Unit   | Catchment 1                                                      | Catchment 2                                       |  |  |
|-------------------------------------------------|--------|------------------------------------------------------------------|---------------------------------------------------|--|--|
| Catchment area at gauge                         | km²    | 38.0                                                             | 9.9                                               |  |  |
| Elevation gradient                              | mASL   | 45-110                                                           | 24-159                                            |  |  |
| Land use distribution                           | -      | Agriculture (73%)<br>Forest (17%)<br>Urban (10%)<br>Pasture (2%) | Agriculture (80%)<br>Pasture (13%)<br>Forest (6%) |  |  |
| Hydrologic Soil Group distribution              | -      | A (19.3%) B (10.8%) C (69.6) D (0.3)                             | A (8.3%)<br>B (7.3%)<br>C (29.1%)<br>D (55.3%)    |  |  |
| Tile drained                                    | %      | 52                                                               | 48                                                |  |  |
| Average annual precipitation (min-max) *        | mm     | 641-809                                                          | 631-945                                           |  |  |
| Average annual maximum temperature (min-max) *  | °C     | 13.1-15.6                                                        | 13.3-15.4                                         |  |  |
| Average annual minimum temperature (min-max) *  | °C     | 4.3-6.1                                                          | 5.6-7.1                                           |  |  |
| Mean runoff rate as percent of precipitation ** | %      | 28-36                                                            | 38-48                                             |  |  |
| Number of subbasins                             | -      | 39                                                               | 17                                                |  |  |
| Number of HRUs                                  | -      | 5163                                                             | 922                                               |  |  |
| Streamflow observation data availability        | mon/yr | 01/1972-04/2024                                                  | 06/2010-12/2013                                   |  |  |
| Metabolite observation data availability        | mon/yr | 10/2016-04/2024<br>(weekly)                                      | 05/2010-12/2013<br>(daily with gaps)              |  |  |
| Metabolite calibration period                   | mon/yr | 10/2016-12/2019                                                  | 06/2010-12/2013                                   |  |  |
| Metabolite validation period                    | mon/yr | 01/2020-04/2024                                                  | Same as calibration period                        |  |  |

<sup>\*</sup> time period Jan-2008 to Dec-2013

<sup>\*\*</sup> time period Jun-2010 to Dec-2013

390 Table 2: Model performance statistics for streamflow and metabolite concentrations in catchments C1 and C2. Performance metrics include Kling-Gupta Efficiency (KGE), correlation coefficient (r), bias ratio (β), and variability ratio (γ). For C1, separate calibration and validation periods were evaluated, while C2 used the complete period. Results are shown as average values from the top 20 parameter sets.

| C1                   |        |      |      | C2                              |      |      |      |      |      |      |
|----------------------|--------|------|------|---------------------------------|------|------|------|------|------|------|
| Complete<br>Period   |        |      |      | Calibration Validation Complete |      |      |      |      |      |      |
|                      | KGE    | r    | ß    | γ                               | KGE  | KGE  | KGE  | r    | ß    | γ    |
| Streamflow           | 0.76   | 0.81 | 1.07 | 0.88                            | -    | -    | 0.78 | 0.79 | 1.02 | 0.97 |
| Metabolite with plan | t 0.67 | 0.7  | 1.03 | 1.11                            | 0.56 | 0.54 |      |      |      |      |
| uptake               |        |      |      |                                 |      |      | 0.46 | 0.74 | 0.61 | 1.27 |
| Metabolite without   | 0.61   | 0.73 | 1.2  | 1.19                            | -    | -    |      |      |      |      |
| plant uptake         |        |      |      |                                 |      |      | 0.50 | 0.75 | 0.66 | 1.28 |

Average values for the 20 best parameterizations

Figure 1: Flow chart of the implemented plant uptake functionality in SWAT

Figure 2: Catchment 1 (C1) time series (Jun 2016 – Oct 2024) for observed and simulated discharge (a), metabolite concentration with plant uptake (b), metabolite concentration without plant uptake (c), and the difference between (c) and (b)

Figure 3: Catchment 2 (C2) time series (Jun 2010 – Dec 2013) for observed and simulated discharge (a), metabolite concentration with plant uptake (b), metabolite concentration without plant uptake (c), and the difference between (c) and (b)

Figure 4: Temporal dynamics of metabolite fate in an agricultural HRU in Catchment 1 showing the relationship between crop rotation, biomass development (Biomass, t/ha), metabolite in soil mass per area (ConcSoil, g/ha), metabolite in plant concentration (ConcPlt, mg/kg), metabolite mass in plants per area (MassPlt, g/ha), and accumulated metabolite plant uptake (AccumMassPlt, g/ha) over a 6-year period (2016-2022). Note the logarithmic scale on the right axis for metabolite in plant concentration and biomass.