# Peer review of "Technical Note: Extending the SWAT2012 and SWAT+ models to simulate pesticide plant uptake processes"

_EGUsphere, 2025_

## Author Comment (AC1)

**Reviewer 2:**

This is a well written manuscript with the objective clearly stated, which is to describe the work extending SWAT to simulate pesticide plant uptake. This addition represents a major update to the SWAT model which has been widely used for water quality research and assessments throughout the world. Descriptions of the model implementation, calibration and evaluation are clear. It is publishable with minor technical corrections as suggested below.

A: Thank you for your positive feedback on our manuscript. We appreciate your recognition that this addition represents a major update to the SWAT model and that the descriptions of the model implementation, calibration, and evaluation are clear. We will address your specific technical corrections in our responses to your detailed comments below.

1. P7, ln205. The model without plant uptake looked under-estimating Catchment 2 data in 2013 onward (Fig 3). How would you justify the need for the plant uptake in this case? Over pre 2013, the model without plant uptake seemed performing generally well (albeit at some level of over-/under- predictions). Improvement wasn't obvious with the case of "with plant uptake" (i.e., comparing Fig 3 Plot b and c). Some more explanation should be useful than simply stating "not all sources of the metabolite were considered".

A: Thank you for this observation regarding the model performance in Catchment 2 (C2). You are correct that the model appears to underestimate metabolite concentrations during this period, and that the implementation of plant uptake does not improve the match with observations in this case.

The underestimation in C2 is due to the fact that, as briefly mentioned in the manuscript (line 185-188), not all sources of the metabolite were considered in our modeling approach. The metabolite we investigated can be formed from multiple parent compounds beyond the single pesticide we simulated. We deliberately chose to model only one parent compound (likely the most significant contributor) as pesticide application data for other parent compounds was not available. This limitation explains the systematic underestimation (bias ratio $\beta$ of 0.66 without plant uptake and 0.61 with plant uptake) observed in C2.

The justification for including plant uptake is not based solely on improving model fit, but rather on incorporating a known and important physical process in pesticide fate modeling.

By implementing this process, we enhance the mechanistic representation of pesticide fate.

Revision planned:

In Section 3.2, we will expand our explanation of the C2 results to clarify why the model underestimates concentrations, provide a few more details about the multiple sources of the metabolite that weren't considered in our simulation, and better explain why plant uptake remains an important process to include despite not visibly improving the match with observations.

2. P7, ln210. Adding plant uptake to the model essentially includes another "sink" term in the mass balance system. There shouldn't be surprises to see reduced concentration predictions. For c2, such reduction seems not needed as comment above. Some data interpretation including potential impact of different factors between c1 and c2 should strengthen the statement in this paragraph. An open discussion otherwise is needed if this result does suggest plant uptake is a catchment- (or crop-) dependent process.

A: Thank you for this comment about the role of plant uptake as a sink term in the mass balance system and the differences observed between catchments C1 and C2.

You are correct that adding plant uptake as a sink term would naturally lead to reduced concentration predictions. Our goal was not to present this as a surprising finding, but rather to quantify the magnitude of this effect and demonstrate how it varies across different catchment conditions.

We further agree that the difference in impact between the two catchments is informative and that the manuscript would benefit from an expanded explanation. As you suggest, the impact of plant uptake on concentration reduction at the catchment scale appears to be influenced by several factors:

1. Land use composition: C1 has 73% agricultural land compared to 80% in C2, but the crop rotations and specific crop types differ between the catchments. C1 has more corn in the rotation, while C2 has more winter cereals. Spring and winter crops have different growing periods and biomass, which affect the water and chemical uptake of the plants.
2. Percent crop treated: The proportion of agricultural land actually treated with the pesticide forming the metabolite varies between catchments, affecting the spatial distribution and total mass of available metabolite for plant uptake.

3. Hydrologic regime: The catchments have different runoff ratios (28-36% in C1 vs. 38-48% in C2 as shown in Table 1). This suggests different partitioning of water between evapotranspiration (which drives plant uptake) and runoff/drainage pathways (which drives off-field transport of the parent compound before the metabolite can form).

4. Metabolite behavior: As noted in our response to your first comment, C2 has multiple sources of the metabolite beyond the simulated parent compound, which affects the relative importance of plant uptake as a sink term.

5. Different pesticide products are used across the catchments, with varying metabolite formation rates. Some products used in C2 may have reduced or no formation of the metabolite, further complicating the mass balance and the relative impact of plant uptake processes.

Revision planned:

In Section 3.2, we will expand our discussion of the factors contributing to different plant uptake impacts between C1 and C2. To this end, we will build upon the catchment-characteristics mentioned above.

3. P15-16, Figs.2 and 3 captions. There appeared to be duplicated "b" denoting Plots b and c.

A: Thank you for pointing out this error in the figure captions. The captions currently state:
"...metabolite concentration with plant uptake (b), metabolite concentration without plant uptake (b), and the difference between (c) and (b)"
This should be corrected to:
"...metabolite concentration with plant uptake (b), metabolite concentration without plant uptake (c), and the difference between (c) and (b)"

Revision planned:

We will correct the captions for Figures 2 and 3 as described above.

---

## Author Comment (AC2)

**Reviewer 1**

This manuscript presents the integration of a function representing the absorption of pesticides by plants in SWAT (version 2012) and SWAT. It then presents its application for a metabolite on 2 catchments. It concludes that it is important to take this process into account, since the simulations carried out lead to decreases in concentration of 5 to 17% compared to simulations that do not take it into account.

The subject is an interesting one, given the growing interest in the role of metabolites in contamination pressure on aquatic environments or on the human supply of drinking water. However, beyond the interest of providing a code integrating this new functionality, it seemed to me that the manuscript would benefit from being more detailed.

A: Thank you for your thoughtful review and for recognizing the importance of this work in the context of metabolite occurrences in river systems. We appreciate your suggestion that the manuscript would benefit from additional details. However, we would like to note that this submission is intended as a technical note rather than a full research article, which constrains the level of detail that can be included while remaining within the format guidelines for HESS technical notes.

The primary purpose of this technical note is to introduce a new plant uptake component to the SWAT modeling community, document its implementation in both SWAT versions, and demonstrate that it functions as intended with expected behaviors. The application examples are provided to illustrate the functionality rather than to draw conclusions about pesticide fate across different environmental settings.

We have carefully considered all your specific comments and have addressed them in our responses below.

1. More details would be appreciated for the description of the soil profile, flow components in it, the formation of metabolites in the profile (for SWAT+) and the way in which the abstraction of water (and pesticides) is modulated according to depth. Are the parameters given for the equations describing the abstraction of water by plants fixed (lines 95 to 105), or can they be varied? Because if it is not possible to modulate the fact that 50% of the water is taken from the top 6% of the soil, pesticides will also be taken as soon as they enter the soil, which may underestimate the risk of infiltration into less biologically active horizons.

A: Thank you for this comment regarding the soil profile representation and water/pesticide processes. You raise a valid point about the potential implications of fixed water uptake

distributions on pesticide fate simulations. We address each aspect of your comment below:

1. Soil profile description and flow components:
   SWAT represents the soil profile through multiple layers with user-defined properties. In the technical note, we referenced the SWAT theoretical documentation (Neitsch et al., 2011) for detailed equations of these processes rather than repeating them. However, we agree that the soil profiles could be better described for an enhanced interpretation of the results, and we plan to revise the manuscript accordingly (see point (1) in planned revisions below).

2. Formation of metabolites in the profile (for SWAT+):
   For metabolite formation in SWAT+, the process occurs within each soil layer via first-order degradation from the parent compound, with the formation fraction defined as a pesticide-specific parameter. This allows metabolites to form at different depths depending on the vertical distribution of the parent compound. The implementation assumes uniform degradation rates across soil depths. We plan to add those missing details to the manuscript (see point (2) in planned revisions below).

3. Water/pesticide abstraction modulation by depth and parameter flexibility:
   The parameters describing water uptake distribution (ubw and uobw in the equation on line 98) are indeed fixed in the current implementation of SWAT. This exponential distribution with depth (where approximately 50% of water uptake occurs in the upper 6% of the root zone) is based on field observations of root density distribution as referenced in the manuscript (Jackson et al., 1996; Feddes et al., 1976). This is the standard implementation in SWAT and was not modified by our extension. We plan to clarify and add a brief discussion regarding this topic to the manuscript (see point (3) in planned revisions below).

4. Concern about underestimating infiltration risk:
   You expressed concern about pesticides being taken up as soon as they enter the soil potentially underestimating infiltration risk to deeper horizons due to the exponential distribution of water uptake mentioned in the point above. While the water uptake pattern is fixed, pesticide movement and availability for uptake is controlled by different processes. The vertical movement of pesticides between soil layers is controlled by soil hydraulic properties and chemical sorption characteristics, which can result in different movement rates compared to water.

Additionally, the implemented plant uptake process only applies to dissolved pesticides in soil pore water, not to pesticides bound to soil particles. We will mention this point and address it by specifically pointing out this limitation (see point (3) in planned revisions below).

Revision planned:
(1) In Section 3 and Table 1, we will add a general description of the soil characteristics in both catchments to provide more context for the model application.

(2) In Section 2.1, we will explicitly explain that in SWAT+, metabolite formation occurs within each soil layer via first-order degradation from the parent compound, with the formation fraction defined as a pesticide-specific parameter. We will clarify that this mechanism allows metabolites to form at different depths depending on the vertical distribution of the parent compound. The change is intended to provide more complete information on how metabolite formation is handled in the different model versions.

(3) In Section 2.2, we will add a note clarifying that the water uptake distribution parameters are fixed in the current SWAT implementations and can only be altered by changing the source code. We will also add a brief statement in Section 4 acknowledging that while this fixed distribution is based on empirical observations, it could potentially affect pesticide fate predictions. We will suggest that future model versions might explore options for user-adjustable water uptake distributions.

2. It is surprising that the parameter describing the uptake of pesticides by plants is linked to the basins.bsn file, when it is explained before that it is a parameter that varies with both the plant and the molecule in question. This choice, which may be constrained by the structure of the model, could have been discussed? In the same way, the fact that it is not possible to represent the return of pesticides to the soil at the time of harvest, when the crop residues are left in place, could have been justified.

A: Thank you for raising these points about parameter implementation and process representation.

1. Implementation of pesticide uptake parameter in basins.bsn file:
   The implementation of the pesticide uptake parameter as a compound-specific parameter in the model (despite the fact that it is also crop dependent) is addressed in Section 2.2 (lines 117-123). We noted that "While pstuptk is theoretically influenced by both plant and compound properties, it is implemented as a compound-specific parameter in the model. This approach reflects the limited availability of plant-specific uptake data." The decision to implement this parameter

in the basins.bsn file in the SWAT2012 version was driven by the model structure which can only fully simulate one pesticide at a time (this pesticide is defined in basins.bsn). Given this limitation, implementing the parameter in basins.bsn (rather than in the pesticide database) was the most practical and user-friendly approach as another pesticide cycling parameter (percop) is also implemented in this file. For SWAT+, which allows processing multiple pesticides at the same time, the implementation was conducted within the pesticide data module. We plan to clarify the technical constraints (see points (1) and (2) in the planned revisions below).

2. Return of pesticides to soil through crop residues:
Regarding ignoring pesticide return to soil through crop residues, we addressed this in Section 2.2 (lines 127-136), where we explained that "Once a chemical is taken up by a plant, it is considered permanently removed from the soil system and does not re-enter the soil through processes like root exudation. This is in line with current research that does not indicate that this pathway significantly contributes to the cycling of pesticides within the soil-plant system (Eze and Amuji, 2024)." At the same time, we acknowledge that the re-entry of pesticides via residual plant parts not removed by harvest is a recognized potential source. However, such re-entry would release residues slowly due to the weathering and decomposition processes of the biomass. We prefer to keep this implementation unchanged because the SWAT plant module cannot explicitly simulate (1) the pesticide release during plant residue decomposition and (2) the in-plant partitioning processes between grain/fruit (which is removed during harvest) and vegetative biomass (which might remain on the field), as discussed in lines 126–130. From our perspective, while implementing pesticide re-entry from crop residues could be a potential model improvement, it would require substantial additional model development that goes beyond the scope of this technical note. We propose to keep this process as a model sink, with the assumption that the user is aware of its limitation (as it is clearly documented in the manuscript) and that it is not a significant exposure pathway according to Eze and Amuji (2024). However, we plan to highlight this component by expanding the discussion in Section 4 (see point (3) in the planned revisions below).

Revision planned:
(1) In Section 2.3, we will clarify the technical constraints that led to implementing the parameter in basins.bsn for the SWAT2012 version.

(2) In Section 4, we will expand the discussion to more explicitly address the potential for future implementations that could accommodate crop-specific uptake factors if such data is available.

(3) In Section 4, we will also mention that future model developments could potentially address the return of pesticides to soil through crop residues at harvest time.

3. The fact of always referring to the PEARL model is also surprising. Whenever PEARL is a model widely used for the approval of substances, the choice of process representation should rather refer to knowledge about the processes rather than to another model (unless it has been validated for the aspects represented, but there is no reference cited on this subject?)

A: We appreciate the reviewer's comment about references to the PEARL model. You rightly point out that our approach should be based on process representation rather than simply adopting the framework of "another model".

We would like to highlight that our manuscript already incorporates process-based knowledge. For example, lines 35-42 discuss the mechanistic understanding of plant uptake pathways (apoplastic, symplastic, transmembrane), and lines 46-52 explain the Plant Uptake Factor (PUF) concept and its physical meaning. Additionally, Section 2.2 describes how pesticide uptake is simulated and elaborates that it is driven by plant water uptake, with a linear relationship between pesticide accumulation and water uptake rates, in agreement with the findings from Lamshoeft et al. (2018).

To strengthen our scientific justification for adopting the conceptual approach of the PEARL model, we propose adding the references below to the manuscript:

1. Leistra et al. (2001), who provided the foundational theoretical framework for plant uptake processes in the PEARL model.
2. Bouraoui et al. (2007), who conducted field validation of pesticide fate predictions from PEARL against real-world data in diverse environmental conditions. The publication confirmed the reliability of the model in predicting pesticide transport and explicitly mentions plant uptake pathways.
3. Jorda et al. (2021), who compared the passive advective uptake approach implemented in PEARL with a mechanistic 3D root model. The authors concluded that the simplified approach from PEARL is effective for regulatory applications while acknowledging limitations in heterogeneous soils.

Revision planned:

In Section 2.2, we will add the references mentioned to demonstrate that our implementation builds upon a scientifically validated approach that aligns with our watershed-scale modeling objectives, where computational efficiency and limited data availability necessitate a simplified but effective representation of plant uptake processes.

Bouraoui F. 2007. Testing the PEARL model in the Netherlands and Sweden. Environmental Modelling & Software. 22(7):937–950. https://doi.org/10.1016/j.envsoft.2006.06.004

Jorda, H., Huber, K., Kunkel, A., Vanderborght, J., Javaux, M., Oberdörster, C., Hammel, K., & Schnepf, A. (2021). Mechanistic modeling of pesticide uptake with a 3D plant architecture model. Environmental Science and Pollution Research, 28, 55678–55689. https://doi.org/10.1007/s11356-021-14878-3 /

Leistra M, Boesten JJTI, van der Linden AMA, Tiktak A, van den Berg F. 2001. PEARL model for pesticide behaviour and emissions in soil-plant systems: Description of the processes in FOCUS PEARL v 1.1.1. Alterra-rapport 013. Wageningen, The Netherlands: Alterra. https://edepot.wur.nl/26563

4. The part dealing with application and validation in the basins is quite frustrating, because it deals with anonymous basins (only one of the two basins is described in the article by Rathjens et al 2023 to which the reader is referred), for an anonymous metabolite emerging from an anonymous substance. Furthermore, it is said that the model has been recalibrated because new data are available, without further details: it would have been useful to clarify this, especially since, depending on the nature of the data, there is no need to recalibrate the hydrological part, but only the part relating to pesticides. Very little is said about the sampling method used (punctual, time-controlled, flow-controlled? ) which nevertheless influences the interpretation that can be made of the data and their use for calibration.

A: We appreciate the reviewer's concerns about the application and validation sections of the manuscript. Your comment raises several important points that we address below.

1. Anonymous catchments and compounds:
   We understand the frustration regarding the anonymization of the catchments and compounds. Unfortunately, this confidentiality is necessary due to business interests and proprietary data agreements between farmers who volunteered pesticide application information and institutions who conducted the monitoring. The manuscript already states that both catchments are located in Western Europe

to provide geographical context without compromising these agreements. We would like to emphasize that the primary purpose of this technical note is to document and demonstrate a new model component (pesticide plant uptake) for the SWAT community rather than to make specific conclusions or management recommendations based on the model results. We intentionally chose a technical note format instead of a research paper. For this purpose, the exact identities of the catchments and compounds are not critical to understanding the implementation and function of the new plant uptake module.

2. Model recalibration:
   Regarding recalibration, we already mention in the manuscript that this was necessary for several reasons (lines 170 to 174): (1) Different metabolite properties compared to the previous study; (2) Version updates in the SWAT model; (3) Newly available longer-term monitoring data for catchment C2 (which was not described in Rathjens et al. 2023); and (4) Implementation of the plant uptake process itself, which changes the overall mass balance. In addition, we would further like to mention that metabolites can function as environmental tracers of hydrological processes, making their behavior sensitive to hydrological parameterization. This connection and newly available information can improve model performance and justify recalibrating both hydrological and chemical parameters. The same applies to SWAT model updates, which can impact hydrological and pesticide processes. We don't plan to add additional information about the (re)calibration procedure because discussing detailed calibration/validation procedures for two catchments is beyond the scope and length limitations of a technical note.

3. Sampling methodology:
   In Table 1 of the current manuscript, we only mention the sampling frequency (weekly for C1 and daily with gaps for C2), but we acknowledge that more details about the sampling techniques would help readers better understand the nature of the data used for calibration and validation (see the planned revisions below).

Revision planned:

In Section 3.1, we will add specific information about the sampling techniques used in both catchments (whether punctual, time-controlled, or flow-controlled) to help readers better understand the nature of the data used for calibration and validation.

5. The authors could have justified the fact that they only worked on concentrations, and not also on pesticide flows. The representation of observations and simulations in double cumulative form (cumulative curves of the quantities of metabolite exported as a function of the cumulative curve of the volumes discharged) would be interesting to better judge the adequacy between data and model. The analysis of a few targeted periods, in addition to or instead of the entire period, would also be more convincing of the fact that it is the integration of uptake by plants that allows for the improvement of the representation of the observed concentrations.

A: We appreciate the reviewer's suggestion to include pesticide mass flows in addition to concentration analyses. Your comment raises several important points that we address below:

1. Focus on concentrations rather than pesticide flows:
   We decided to focus primarily on concentration data for the following reasons:
   (1) The available monitoring data was collected in weekly samples for catchment C1 and "daily with gaps" for C2 (as noted in Table 1). The temporal resolution of C1 is insufficient for accurate calculation of continuous pesticide mass loads, particularly during runoff events when concentrations can change rapidly. The interpolation required between sparse concentration measurements would introduce substantial uncertainty in mass flow estimates that would limit their interpretive value.
   (2) As noted in our Results section, we observed water balance discrepancies between gauging stations, particularly during low-flow conditions. These uncertainties in the hydrological measurements would propagate into mass flow calculations, compounding the concentration interpolation errors mentioned above.
   (3) The plant uptake process we are examining involves storage components that change over time (soil retention, groundwater storage, plant tissue accumulation). A simple cumulative export analysis wouldn't fully capture these dynamics, as there are significant time lags between processes (application, degradation, plant uptake, and transport).

2. Suggestion for double cumulative analysis:
   We acknowledge the potential value of double cumulative analysis (plotting cumulative metabolite export versus cumulative discharge) as this approach is occasionally used in hydrological studies for rainfall-runoff relationships. However, this approach is less common for pesticide transport analysis due to the practical limitations mentioned above. We believe our current approach using concentration

time series and HRU-level storage analysis (Figure 4) effectively demonstrates the impact of plant uptake processes on model predictions, which was the primary goal of this technical note.

3.  Analysis of targeted periods:
    We believe that examining the entire simulation period is necessary for this particular process due to the delayed metabolite formation, storage dynamics of metabolites in soil and plants. Plant uptake interacts with soil-hydrologic processes and can have significant time lags (e.g., due to dry and wet years) and seasonal dependencies that require analysis ideally over multi-year periods.

Revision planned:

We do not plan to revise the paper based on this comment as we believe the current analysis appropriately addresses the core scientific questions relevant to this technical note.

6.  In fact, the spatial and temporal scales used to present the application of the developed model are not the most suitable for convincing of the merits of representing plant uptake: at the scale of a 10 or 38 km$^2$ catchment, PUF can be seen as an additional calibration parameter... Performing an analysis at a more restricted scale would have been appreciated.

A: We appreciate the reviewer's concern about the spatial scale of our analysis. Your comment raises two important points about the appropriate scale for demonstrating the plant uptake process implementation that we address below:

1.  Concerns about spatial and temporal scales used in the analysis:
    The catchment scales used in our study (9.9 km$^2$ and 38 km$^2$) are typical and appropriate scales for SWAT watershed modeling applications. While we understand the reviewer's desire and general need for analysis at more restricted spatial and temporal scales, we have already incorporated this through our HRU-level analysis. Another limitation is measurement data availability as observed data was only available at these spatial scales. Figure 4 specifically presents the detailed dynamics of metabolite fate in an agricultural HRU, showing the relationship between crop rotation, biomass development, metabolite concentrations in soil and plants, and accumulated plant uptake over a 6-year period.

2. Concerns about PUF being used as a calibration parameter:
We fully agree with the reviewer that the Plant Uptake Factor should not be seen as a calibration parameter. We understand the concern that at larger scales, the Plant Uptake Factor (PUF) could be viewed as an additional calibration parameter and plan to revise the manuscript accordingly (see planned revisions below).

Revision planned:

In Section 3.1, we will modify the manuscript to strongly recommend that the Plant Uptake Factor should not be used as a calibration parameter but should be derived from laboratory studies or literature values to maintain the physical meaning of the parameter and avoid compensating for other model deficiencies.

7. In Figure 1, it seems that the new routine also modifies the representation of the flow in the drains: yet plants do not withdraw water in this drain section?

A: Great observation. We appreciate the reviewer's careful examination of Figure 1. Rooting depth can extend below tile drains in the SWAT model. However, tiles are only active when soil water content exceeds field capacity, and once water reaches the tile drains, it is not available for plant uptake. The uptake of water and pesticides by plants only occurs from the soil matrix within the rooting zone, not from the drainage system itself.

Revision planned: Figure 1 will be revised by removing the tile drain component from the diagram, as it is not directly relevant to the plant uptake process discussed and could lead to misinterpretation. We might try to add a plant with its root to the figure for illustrating that the rooting depth limits plant uptake.

---

## Author Response (AR2)

**Reviewer 1**

The authors responded thoroughly and convincingly to almost all of the comments addressed to them. The additions to the manuscript clarify some details of the implementation of pesticide plant uptake in SWAT2012 and SWAT+, and explicitly state the constraints and limits related to the models' structure.

Some additional details were added to provide a clearer picture of the two catchments on which the model was tested. Following a reviewer's comment, Figure 1 was modified so that the drainage compartment is no longer visible: given the high proportion of drainage in the application catchments, this does not appear to be the best solution. This figure should be modified further (or a few sentences should be added to the text) to make the indirect influence of plant uptake on drainage fluxes clearer.

A: Thank you for the positive evaluation of our review efforts. We agree that without the tile drain, the hydrological processes are not correctly represented. We have added the tile drain back into the figure but made sure that no pesticide uptake occurs directly from the drain.

**Modified Figure:**